

# Modeling of terrain effect in magnetotelluric data from Garhwal Himalaya Region

Suman Saini[1*], Deepak Kumar Tyagi[2], Sushil kumar[3], Rajeev Sehrawat[1**]

[1]Department of Physics, M. M. Engineering College, Maharishi Markandeshwar (Deemed to be) University, Mullana-Ambala, Haryana, India 133207

Corresponding Author- Rajeev Sehrawat[1**]

 **Email: rajeev.sehrawat@mmumullana.org

*Email: sumanabcd12@gmail.com

[2]Department of Physics, Krishnan College of Science and IT, M.J.P. Rohilkhand University, Bijnor, Uttar Pradesh, India-246701

[3]Department of Geophysics, Kurukshetra University, Kurukshetra, India-136119

## **ABSTRACT**

The magnetotelluric method (MT) is one of the most effective geophysical techniques for studying the deep structure of the Earth's crust, particularly in steep terrain like the Garhwal Himalaya region. The MT responses are distorted as a result of the undulated/rugged terrain. Such responses, if not corrected, can lead to a misinterpretation of MT data for the geoelectrical structures. In this research paper, two different correction procedures were used to compute the topography distortion for the synthetic model of Garhwal Himalaya region from Roorkee to Gangotri section. A finite difference algorithm was used to compute MT responses (apparent resistivity and phase) for the irregular terrain. The accuracy of the terrain correction procedures was checked on results published in the literature on different topography models at various periods. The relative errors between flat earth response (FER) and two terrain correction procedures (TCR1 and TCR2) were calculated and were very less or almost zero for most of the sites along the Roorkee to Gangotri profile except at the foothill where the error was high at lower periods. The similar topography response, terrain corrected responses TCR1 and





TCR2 responses concluded that there is no need for topography correction along Roorkee-Gangotri
Profile because the slope angle is less than one degree.
**Keyword**s: Magnetotelluric, Topography correction procedures, Himalaya region
1. **INTRODUCTION**
The magnetotelluric (MT) method was first explored by Tikhonov (1950) and Cagniard (1953) and
was used to analyse the time-varying measured components of earth's natural time-varying electric
and magnetic fields to determine the interior of the earth. MT technique has been successfully used to
explore a variety of earth resources, including oil, gas, mineral, and geothermal energy (Zhang et al.,
2014; Patro et al., 2017; Mohan et al., 2017).  The MT method is effective for analysing deep crystal
structures in challenging undulating terrains, such as the Himalayan region as compared to the seismic
method (Tyagi , 2007; Israil et al., 2008, 2016; Pavan Kumar et al., 2014; Patro and Harinarayana,
2009; Kumar et al., 2018, 2022; Xiong-Bin, 2020; Dharmendra Kumar et al., 2021; Konda et al.,
2023). Topography affects both the electric field and magnetic field components due to undulating
topographical features like hills and valleys, which distort the current lines (Wannamaker et al., 1986;
Michel Choutraus et al., 1988; Changhong et al., 2018; Kumar et al., 2018, 2022). Therefore, the MT
response functions impedance and apparent resistivity get distorted when the MT sites are on or near
the top of the hill or close to the valley.
Analytical and numerical techniques have been used to measure the topography distortion effect from
MT data. Analytical techniques based on conformal mapping were used by Thayer (1975),
Harinarayana and Sarma (1982). Numerical techniques have been used for different types of terrain
geometrics to remove topography effects from the data (Wannamaker et al., 1986; Michel and
Bouchard, 1988; WescotHessler, 1962; Faradzhev, 1972). The distortions in MT data due to





topography and near-surface inhomogeneities have been observed by many researchers (Jiracek, 1990;
Vozoff, 1991). The distortion tensor stripping-off technique has been used to reduce the topographic
effect and to remove the distortion due to the near-surface heterogeneity (Larsen, 1971). The
analogue, analytic, and numerical solution methods were used to study the analogue model (Wescott
and Hessler, 1962; Faradzhev et al., 1972).Various two-dimensional (2D) numerical techniques have
been used for the numerical treatment of the topographic effects like networking analogy (Ku et al.,
1973; NgCo, 1980) and Rayleigh scattering numerical modeling techniques (Jiracek Redding and
Kojima, 1989)  and finite element method (Wannamaker Stodt and Rijo, 1986; Frankle et al., 2007).
In 2D, the topography effect is galvanic in Transverse Magnetic (TM) mode and inductive in
Transverse Electric (TE) mode, hence more distortion in TM mode than TE mode (Gurer and Ilikisik,
1997; Kumar et al., 2014;  Kumar et al., 2018, 22).
In this study, modified 2D forward and inversion modeling code EM2INV (Rastogi, 1997) based on
the finite difference method were used to compute MT forward modeling responses over flat earth and
topographic surface. Two different terrain correction procedures have been used in this study, first
correction procedure was adopted from Chouteau and Bouchard (1988) and the second was adopted
from Nam et al., (2008) to compute the topography distortion for the synthetic model of Garhwal
Himalayan region (Roorkee-Gangotri section). The results of both terrain correction procedures have
been compared to the model used by Chouteau and Bouchard (1988).
**2. METHODOLOGY**
The topography correction to the MT data has been applied by two different techniques. The first
technique was introduced by Chouteau and Bouchard (1988) to estimate the distortion tensor and
correction of MT data before inversion of MT data. In the second approach, the distortion tensor



stripping-off technique was used to remove the distortion from the MT data (Larsen, 1977 and Nam et
al., 2008). Two correction procedures first adopted by Chouteau and Bouchard (1988) and second by
Nam et al., (2008) were used to correct the MT data.
**2.1 Terrain correction procedure 1 (TCP1):-**
The computational algorithm for 2D forward modeling has been used to account for irregular terrain.
The distortion tensor for the topographic effect was calculated using the technique adopted by
Chouteau and Bouchard (1988). Based on the assumption that the topography distorted subsurface
field can be approximated by multiplying the distortion tensor by the subsurface field for a flat earth
given by:
$\widetilde{E_D} = D\widetilde{E_N}$                                                                         (1)
Were $\widetilde{E_D}$  and $\widetilde{E_N}$ are the distorted and normal electric field matrices with elements $E(f,r)_D$ and
$E(f,r)_N$ respectively. $\widetilde{D}$ is the distortion tensor with elements $D(f,r)$, where $f$ is frequency and $r$ is
the measuring site position. In case of 2D problem in H- polarization and x-axis is the direction of
strike, equation (1) can be written as
$E_{XD}(f,r) = D_{XX}(f,r)E_{XN}(f,r)$                                                      (2)
The impedance tensor can be calculated by dividing equation (2) with magnetic field $H_Y$.
$Z_D(f,x) = D(f,x)Z_N(f,x)$                                                                (3)
Where $Z_N(f,x)$ and $Z_D(f,x)$ are respectively the normal (flat earth) impedance and distortion
impedance. Distortion coefficients $D(f,x)$ are complex coefficients that should just reflect
topography effect. The distortion coefficients are calculated by normalizing the impedances $Z_t(f,x)$



computed over topographic model above a homogeneous medium with the half-space impedance.
Thus, the corrected impedance over flat earth can be calculated by taking the following ratio of the
observed impedances, $Z_D(f, x)$, over irregular topography to the distortion coefficients $D(f, x)$:
$$Z_C(f, x) = Z_D(f, x)/D(f, x) \qquad (4)$$
Where $Z_C(f, x)$ is terrain-corrected impedance.

**2.2 Terrain correction procedure 2 (TCP2):-**

In this correction procedure, the MT data was corrected using the technique adopted by Nam et al.,
(2008). Larsen (1977) introduced the distortion tensor stripping-off technique, in which the
undistorted impedance tensor can be calculated using a linear relationship between the distorted and
undistorted impedance tensor, and topography distorted MT data can be corrected by computing the
distortion tensor. The undistorted impedance tensor is linearly related to the distorted impedance
tensor as:
$$Z^D = D^Z . Z^U \qquad (5)$$
Where $Z^D$ is the distortion impedance tensor, $D^Z$ is distortion tensor and $Z^U$ is the undistorted
impedance tensor respectively. The distortion tensor can be calculated from the relation between the
impedance tensor for a homogeneous medium with topography earth surface ($Z^t$), and that with the
flat earth surface ($Z^h$) as
$$Z^t = D^Z . Z^h \qquad (6)$$
In case of 2D, $Z^h_{xx} = Z^h_{yy} = (0, 0)$ and $Z^h_{xy} \neq -Z^h_{yx}$, the inhomogeneous earth distortion tensor,
equations (5) and (6) can be rewritten in matrix form as



$$\begin{bmatrix} 0 & Z_{xy}^D \\ Z_{yx}^D & 0 \end{bmatrix}\begin{bmatrix} 0 & D_{xy}^Z \\ D_{yx}^Z & 0 \end{bmatrix}\begin{bmatrix} 0 & Z_{xy}^U \\ Z_{yx}^U & 0 \end{bmatrix}$$
(7)

and
$$\begin{bmatrix} 0 & Z_{xy}^t \\ Z_{yx}^t & 0 \end{bmatrix} = \begin{bmatrix} 0 & D_{xy}^Z \\ D_{yx}^Z & 0 \end{bmatrix}\begin{bmatrix} 0 & Z_{xy}^h \\ Z_{yx}^h & 0 \end{bmatrix}$$
(8)

So
$$\begin{bmatrix} 0 & D_{xy}^Z \\ D_{yx}^Z & 0 \end{bmatrix} = \begin{bmatrix} 0 & Z_{xy}^t \\ Z_{yx}^t & 0 \end{bmatrix}\begin{bmatrix} 0 & Z_{xy}^h \\ Z_{yx}^h & 0 \end{bmatrix}^{-1}$$
(9)

$$\begin{bmatrix} 0 & D_{xy}^Z \\ D_{yx}^Z & 0 \end{bmatrix} = \begin{bmatrix} (Z_{xy}^t)/(Z_{xy}^h) & 0 \\ 0 & (-Z_{yx}^t)/(Z_{yx}^h) \end{bmatrix}$$
(10)

Substituting equation (10) in equation (7)
$$\begin{bmatrix} 0 & Z_{xy}^D \\ Z_{yx}^D & 0 \end{bmatrix} = \begin{bmatrix} (Z_{xy}^t)/(Z_{xy}^h) & 0 \\ 0 & (-Z_{yx}^t)/(Z_{yx}^h) \end{bmatrix}\begin{bmatrix} 0 & Z_{xy}^U \\ Z_{yx}^U & 0 \end{bmatrix}$$
(11)

The  undistorted or corrected impedance tensor component can be obtained as
So
$Z_{xy}^U = (Z_{xy}^h Z_{xy}^D)/(Z_{xy}^t)$
(12)

$Z_{yx}^U = (Z_{yx}^h Z_{yx}^D)/(Z_{yx}^t)$
(13)

## 120   3. TESTING THE CORRECTION PROCEDURES:

In this study, we replicated the model of Chouteau and Bouchard (1988). A 2D topographic
homogeneous model of 500 Ω-m half-space with a resistive block of 10000 Ω-m having a thickness of
1 km was embedded in the model from surface relief (Fig. 1). The MT responses for the model have
been computed with and without topography. The terrain correction procedures (TCP1 & TCP2) have



been applied to the model responses at a particular period of 0.1 second (sec) and validated over the
inhomogeneous model of Chouteau and Bouchard (1988). In 2D the topography effect is galvanic in
TM mode and inductive in TE mode. Therefore, the comparison of TM component of flat earth
response (FER), topographic response (TR) and two terrain correction responses (TCR1 and TCR2)
were shown in Fig. 2. It is concluded from the Fig. 2 that the TCR1 and TCR2 are very similar to the
FER at particular period of 0.1 sec, but not similar to the TR, which shows a good agreement of
published result of Chouteau and Bouchard (1988).

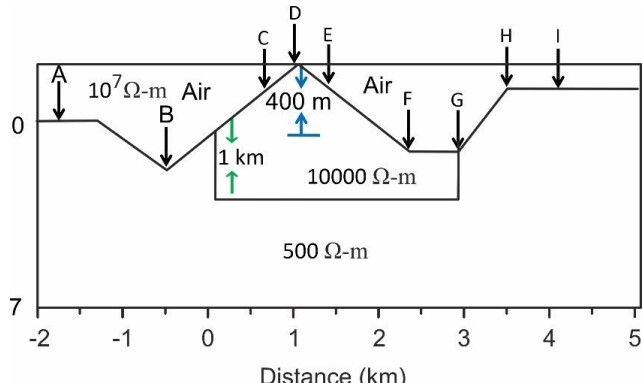


**Fig. 1:** Topographic model of 500 Ω-m half-space with a resistive body of 10000 Ω-m was embedded
from the surface relief (Chouteau and Bouchard, 1988).

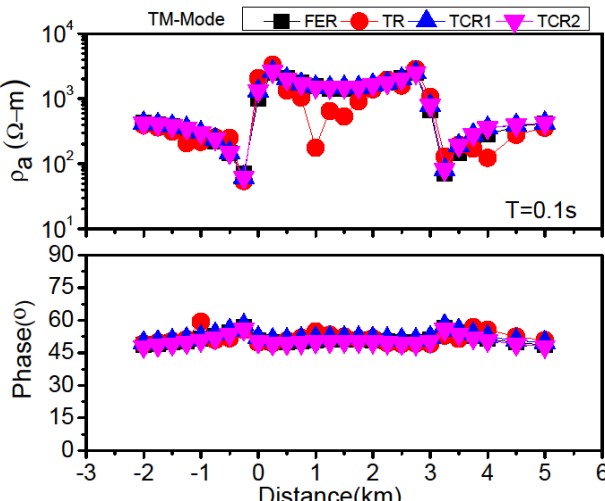


**Fig. 2:** Comparison of TM component of flat earth response (FER), topographic response (TR) and

two terrain correction responses (TCR1 and TCR2) at 0.1 sec.

Fig. 3 showed that the topography distortions are large for higher period in apparent resistivity

component only, which shows the galvanic nature of the topography distortions. The terrain corrected

responses (TCR1 and TCR2) in Fig. 3 are almost similar to flat earth responses (FER) at six periods

(0.001 sec, 0.01 sec, 0.1 sec, 1 sec, 10 sec, and 100 sec respectively). Relative errors were also

calculated to check the accuracy of the terrain correction responses (TCR1 and TCR2) with flat earth

responses at these periods. The relative error between the FER and TCR1 and TCR2 were very small

at all these periods except at site D only at lower periods (because of 10000 $\Omega$-m resistive body) as

shown in Fig. 4. This shows the accuracy of the correction procedures.



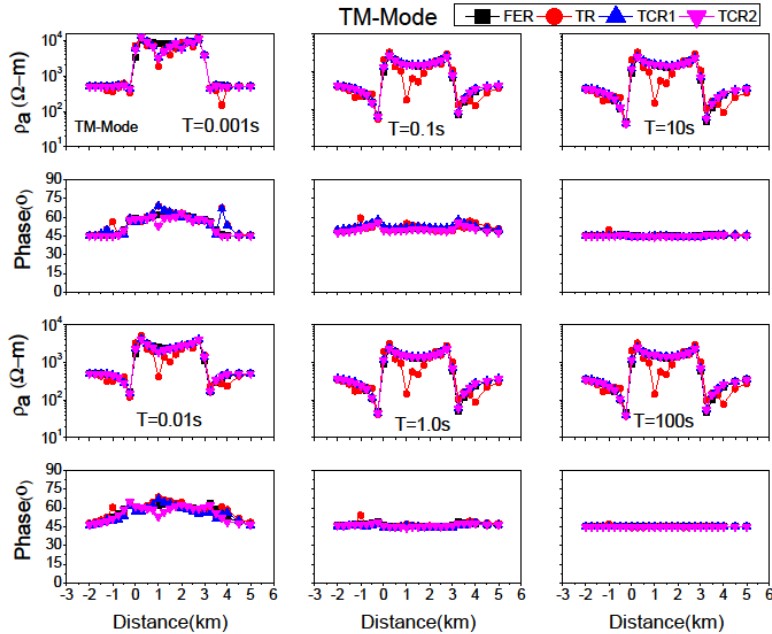

**Fig. 3:** Comparison of TM components of flat earth response (FER), topographic response (TR), and two correction procedures (TCR1 and TCR2) for the model in Fig. 1 at six different periods (0.001 sec, 0.01 sec, 0.1 sec, 1sec, 10 sec and 100 sec).



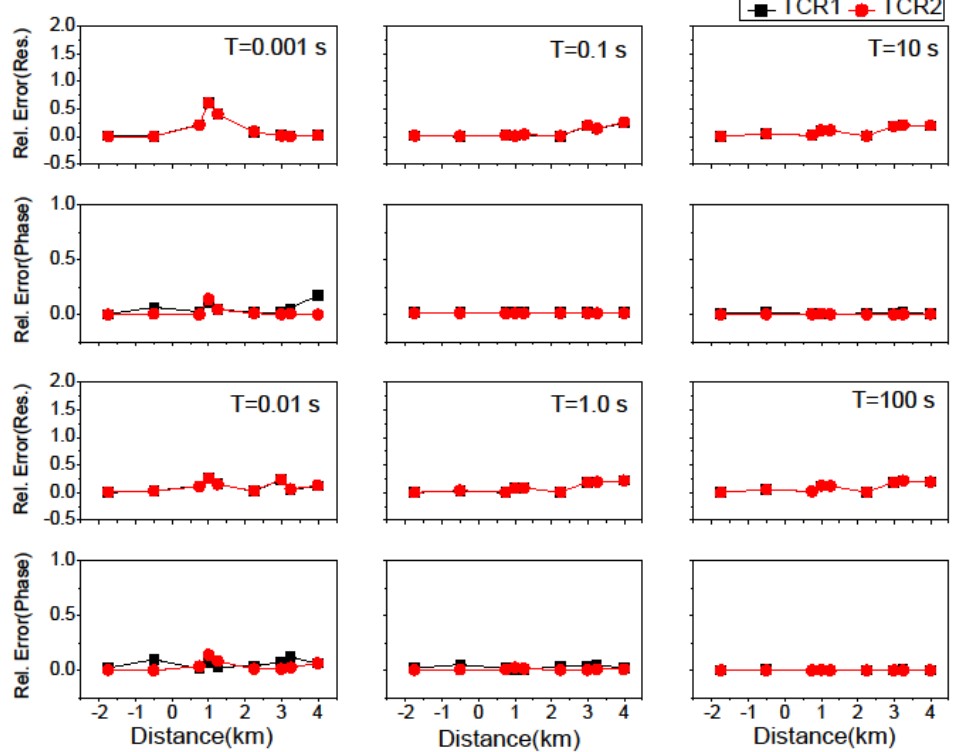

150

**Fig. 4:** Relative error between terrains corrected responses (TCR1 and TCR2) with respect to flat

earth responses (apparent resistivity and phase) at six different periods with homogeneous half-space

of 500 Ω-m resistivity.

## 4. MODELING OF ROORKEE TO GANGOTRI SECTION:

A theoretical analysis of the effect of topography on MT responses was also taken into account in the

Himalayan topography model. A theoretical model of Roorkee-Gangotri Profile was generated to

simulate the MT response. To compute the MT forward modeling responses over the rugged

topographic surface in Roorkee to Gangotri section, the input model was prepared from a 2D inverted

geoelectrical resistivity model (Tyagi, 2007).The topography model having an elevation of 2.75 km



consists of a 180 km long profile from Roorkee to Gangotri drawn (Tyagi, 2007; Suman et al.,
2023).In this model, two conductive blocks having resistivity 30 Ω-m and 10 Ω-m were embedded in
a homogeneous half-space of 100 Ω-m resistivity. The first block of resistivity 30 Ω-m having width
80 km and thickness 6 km was embedded just near the earth's surface relief and the second block of
width 40 km and thickness 25 km was embedded at 6 km depth from the surface. The MT responses
were computed by considering three models, (1) one with half-space of resistivity 100 Ω-m (Fig. 5a),
(2) with half-space of resistivity 500 Ω-m, (3) with an additional resistive body of 8000 Ω-m
embedded from earth surface relief having thickness about 6 km with half-space of resistivity 100 Ω-
m as shown in Fig. 5b. The topography response (TR), flat earth response (FER) and two topography
corrected responses (TCR1 &TCR2) were analysed for nine sites (A, B, C, D, E, F, G, H& I) as
shown in Fig. 5 at six distinct periods (0.00131 sec, 0.0102 sec, 0.1063 sec, 1.1110 sec, 11.6078 sec,
and 121.2813 sec).

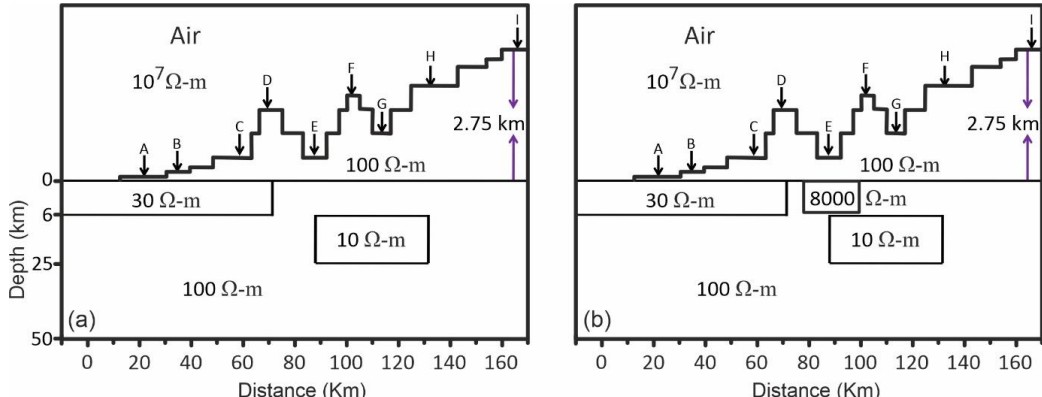


**Fig. 5:** (a) A Synthetic model of Garhwal Himalaya along Roorkee to Gangotri Profile in half-space of
resistivity 100 Ω-m (b) with a resistive block of resistivity 8000 Ω-m.
**5. RESULT AND DISCUSSION:**

**5.1. Model with half-space of resistivity 100Ω-m:**



The topography response (TR) and flat earth response (FER) were computed for the topography
model with a conducive body of 30 Ω-m resistivity in a half-space of 100 Ω-m resistivity (Fig. 5a) and
the topography corrections procedures were applied to the MT data. Fig. 6 shows the TM mode of
topography response (TR), flat earth response (FER) and two topography correction responses (TCR1
& TCR2) at six different periods (0.00131 sec, 0.0102 sec, 0.1063 sec, 1.1110 sec, 11.6078 sec, and
121.2813 sec). The topography effect depends upon the ramp/slope angle of the hill and is significant
when the slope angle is greater than 7.5º (Kumar et al., 2018). It is clear from Fig. 6 that the TCR1
and TCR2 are almost similar to the topographic response, because the slope angle is less than 1º. The
TCR1 &TCR2 were not similar to the flat earth response for the sites from A to D at lower periods
0.00131 sec, 0.0102 sec, 0.1063 sec and 1.111 sec, because of the exposure of the conductive body
having resistivity 30 Ω-m to the surface (from A to D) and its galvanic effect. The relative errors were
also calculated between the FER with TCR1 and TCR2 and were high for the sites A, B and C for
lower periods (0.00131sec, 0.0102 sec and 0.10631 sec) due to the presence of the conductive body
underneath these sites and was very small for all other sites D, E, F, G, H and I at all periods as shown
in Fig. 7.

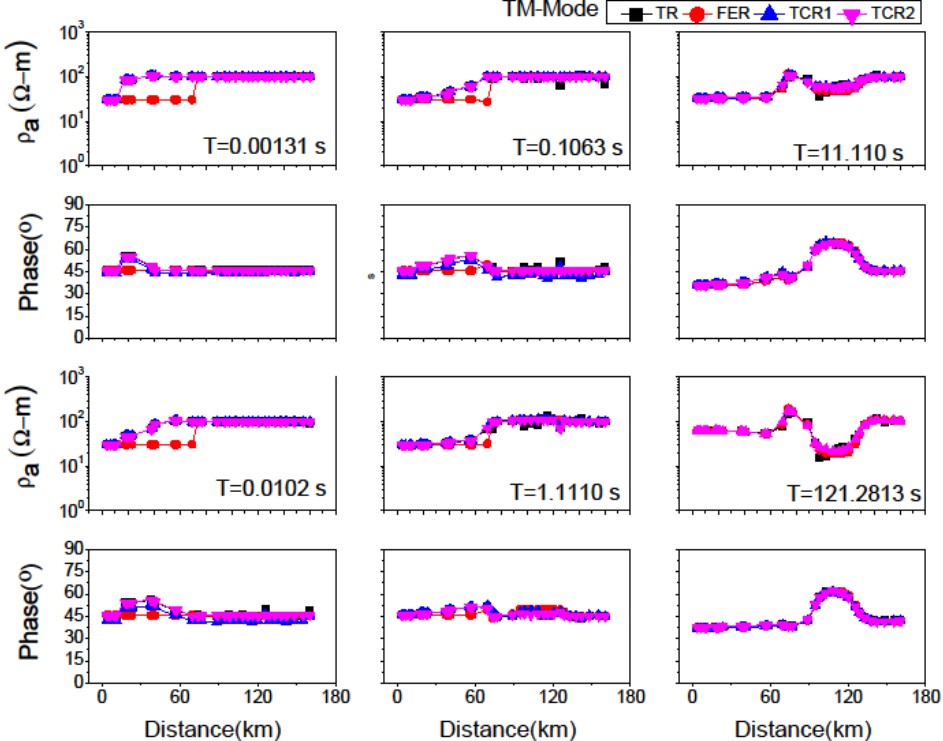


**Fig. 6:** Comparison of TM components of flat earth response (FER), topographic response (TR), and two correction procedures (TCR1 and TCR2) at six different periods for homogeneous half-space of resistivity100 Ω-m.



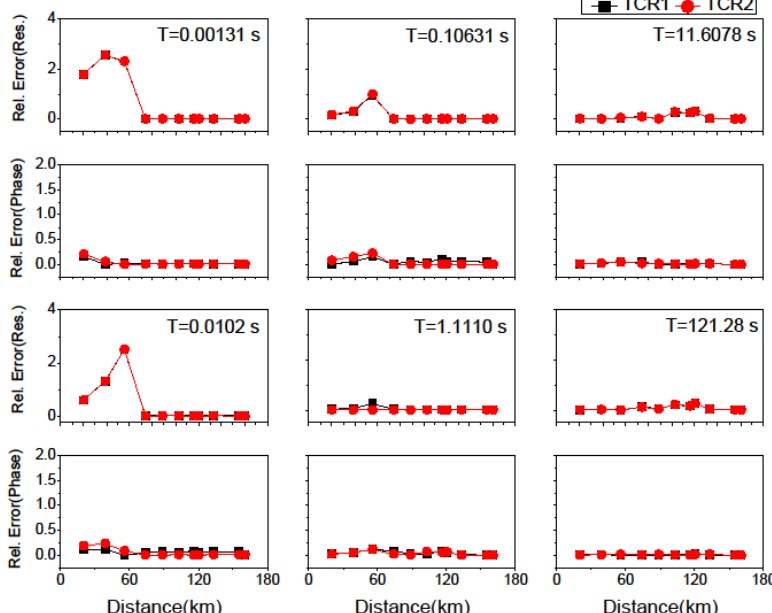


**Fig. 7:** Relative error between terrains corrected responses (TCR1 and TCR2) with respect to flat
earth response (apparent resistivity and phase) at six different periods with half-space of resistivity
100 Ω-m.

### 5.2. Model with half-space of resistivity 500Ω-m:

Now consider the case in which model half-space resistivity was replaced with 500 Ω-m in Fig. 5a.
The topography response (TR) and flat earth response (FER) were computed for the topography
model with half-space of 500 Ω-m resistivity (Fig. 5a) and the topography correction procedures were
applied to the MT data. Fig. 8 shows the TM component of topography response (TR), flat earth
response (FER), and topography corrected responses (TCR1 & TCR 2) at six different periods. The
results were almost similar to the response of the model with half-space of resistivity 100 Ω-m. The
relative errors were also calculated in this case also between the FER with TCR1 and TCR2 and the
results were similar to the model with half-space of 100 Ω-m at all these periods (0.00131 sec, 0.0102
sec, 0.1063 sec, 1.1110 sec, 11.6078 sec, and 121.2813 sec) as shown in Fig. 9.



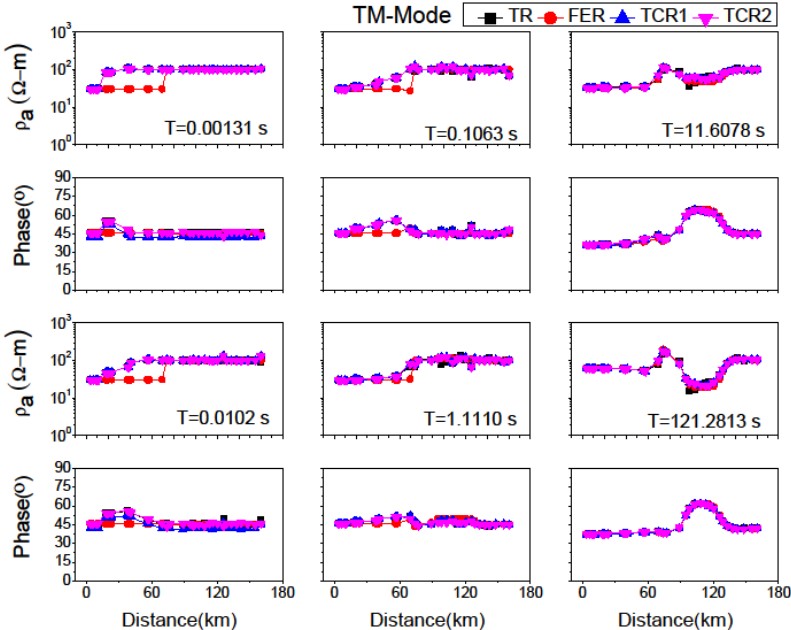

**Fig. 8:** Comparison of TM components of flat earth response (FER), topographic response (TR), and

two correction procedures (TCR1 and TCR2) at six different periods for half-space of resistivity 500

Ω-m.



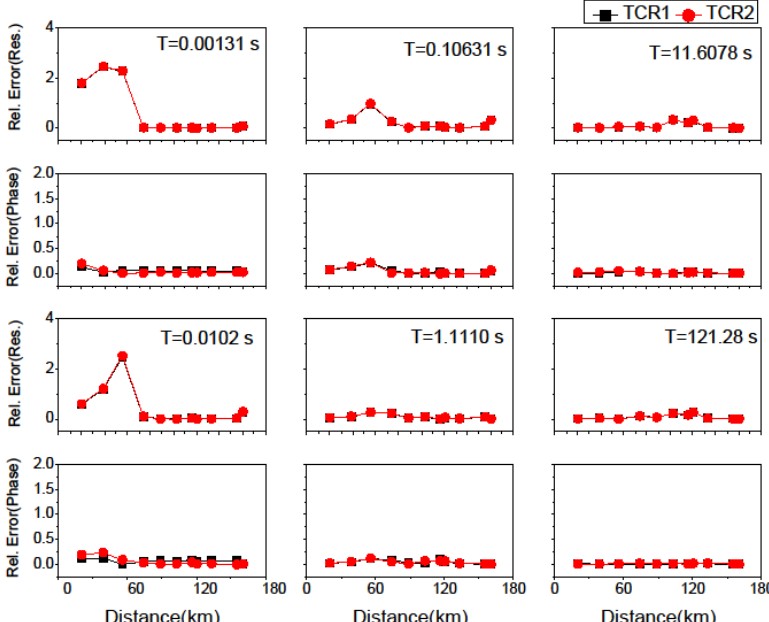

**Fig. 9:** Relative error between terrains corrected responses (TCR1 and TCR2) with respect to flat

earth response (apparent resistivity and phase) at six different periods with half-space of resistivity

500 Ω-m.

### 5.3. Model with a resistive block of resistivity 8000 Ω-m in half-space of 100Ω-m:

The topography response (TR) and flat earth response (FER) were also computed for the topography

model with a resistive block of resistivity 8000 Ω-m in half-space of 100 Ω-m resistivity (Fig. 5b) and

the topography corrections were applied to the MT data. Fig. 10 shows the TM component of

topography response (TR), flat earth response (FER) and two topography correction responses (TCR1

& TCR2) at six different periods. The TCR1 & TCR2 were not similar to the flat earth model for the

sites from A to F, because of the exposure of the conductive body having resistivity 30 Ω-m to the

surface (from A to D) and its galvanic effect and the presence of 8000 Ω-m resistive body (from D to



F). The relative errors were also calculated between the FER with TCR1 and TCR2 and were high for
the sites A, B and C for lower periods (0.00131 sec, 0.0102 sec and 0.10631 sec) due to the presence
of the conductive body underneath these sites and for higher periods (1.1110 sec, 11.6078 sec and
121.2813 sec) the relative error was again high due the presence of the 8000 Ω-m resistive body from
D to F as shown in Fig. 11.

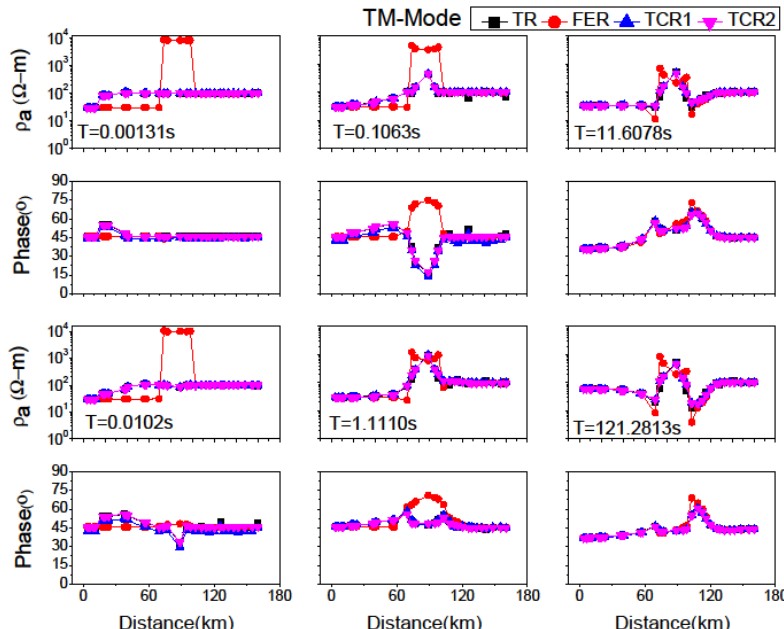


**Fig. 10:** Comparison of TM components of flat earth response (FER), topographic response (TR), and
two correction procedures (TCR1 and TCR2) at six different periods for half-space of resistivity 100
Ω-m.



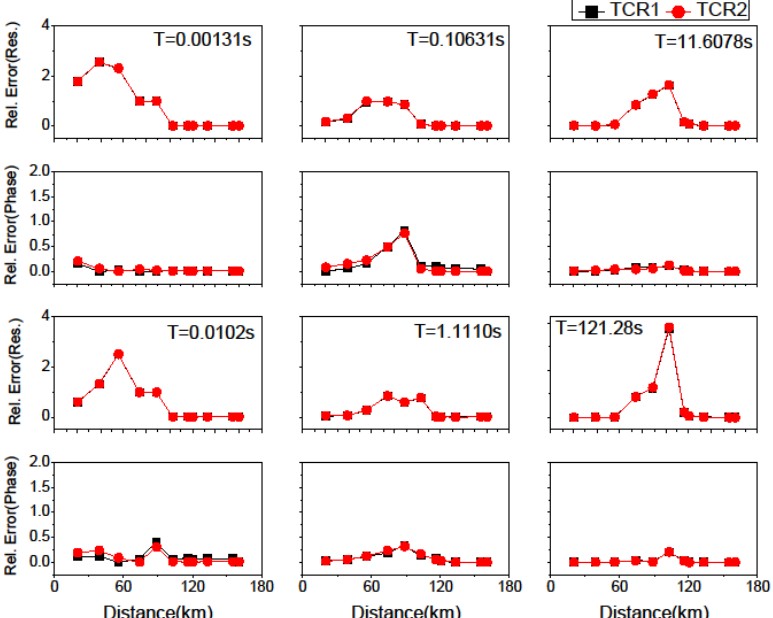


**Fig. 11:** Relative error between terrains corrected responses (TCR1 and TCR2) with respect to flat

earth response (apparent resistivity and phase) at six different periods with half-space of resistivity

100 Ω-m.

## 6. CONCLUSIONS:

The study shows the effect of topography in the MT data along a synthetic model of Roorkee-

Gangotri profile.The two correction procedures were used to remove the topography distotion from

MT data. The similar FER, TCR1 and TCR2 in Fig. 3 shows that the both correction procedures are

capable to remove the topography effect, this shows the afficacy of the two correction procedures. The

similar TR, TCR1and TCR2 responses (Fig. 6, 8 and 10) concluded that there is no need for

topography correction along Roorkee-Gangotri Profile, because the slope angle is less than one

degree. The relative error between the FER and TCR1 and TCR2 also showed the accuracy of the two



correction procedures (TCR1 & TCR2) in this study. The presence of near surface
hetrogeneity/surface exposure of conductive/resistive body also distort the MT responses as in this
model (the FER not similar to TR, TCR1 and TCR2).
**ACKNOWLEDGEMENT**
Suman thank Head of Department, Department of Physics, M. M. Engineering College, Maharishi
Markandeshwar (Deemed to be) University for continuous guidance and support throughout this
research.
**AUTHOR CONTRIBUTION:**
Dr. Deepak Kumar Tyagi and Ms Suman Saini designed the experiments, developed the model
and performed the simulations. Dr Rajeev Sehrawat, Dr. Sushil Kumar prepared the manuscript
with contributions from all the co-authors.
**COMPETING INTERESTS**
The authors declare that they have no conflict of interest.
**REFRENCES**
1. Cagniard, L.: Basic theory of the magneto-telluric method of geophysical prospecting.
Geophysics. **18** 605-635, 1953.
2. Changhong, Lin.: The effects of 3D topography on controlled-source audio-frequency
magnetotelluric responses. Geophysics. **83** no. 2 p. Wb97–wb108, 1910.1190/geo-0429, 2018.
3. Chouteau, M., and Bouchard, K.: Two-dimensional terrain correction in magnetotelluric. Surveys
Geophysics. **53** 854-862, 1988.
4. Coggon, H.: Electromagnetic and electrical modeling by the finite-element method: Geophysics.
**36** 132-155, 1971.



5.  Faradzhev, A. S., Kakhramanov, K. K., Sarkisov, G. A., and Khalilova, N. E.: On effects of
terrain on magnetotelluric sounding (MTS) and profiling (MTP). Izvestia Earth Physics. **5** 329-
330, 1972.

6.  Franke, A., Börner, R. U., and Spitzer, K.: Adaptive unstructured grid finite element simulation of
two-dimensional magnetotelluric fields for arbitrary surface and seafloor topography.
Geophysical Journal International. **171** 71-86, 2007.
7.  Gurer, A., and Ilkisik, M.: The importance of topographic corrections on magnetotelluric response
data from rugged regions of Anatolia, Geophys. Prospect. **45** 111-125, 1997.
8.  Harinarayana, T., and Sarma, S. V. S.: Topographic effects on telluric field measurements.
Pageoph. **120** 778-783, 1982.
9.  Holcombeand, H. T., and Jiracek, G. R.: 1984 Three-dimensional terrain corrections in resistivity
surveys: Geophysics. **49** 439-452, 1977.
10. Israil, M., Tyagi, D. K., Gupta, P. K., and Niwas, Sri.: Magnetotelluric investigations for imaging
electrical structure of Garhwal Himalaya corridor, Uttarakhand, India. Journal of Earth System
Sci.**117** 189-200, 2008.
11. Israil, M., Mamoriya, P., Gupta, P. K., and Varshney, S. K.: Transverse Tectonics Feature
Delineated by Modelling of Magnetotelluric Data from Garhwal Himalaya Corridor, India. Curr.
Sci. **111** 868-875, 2016.
12. Jiracek, G. R., Redding, R. P., and Kojima, R. K.: Application of the Rayleigh-FFT techniqueto
magnetotelluric modeling and correction. Physics of the Earth and Planetary Interiors, **53** 365-
375, 1989.

13. Jiracek, G.: Near-surface and topographic distortions in electromagnetic induction. Surveys
Geophysics. **11** 163-203, 1990.
14. Konda, S., Patro, P. K., Reddy, K. C., and Babu, N.: Three-dimensional magnetotelluric
signatures and rheology of subducting continental crust: Insights from Sikkim Himalaya, India.
Journal of Geodynamics. **155** 101961, 2023.
15. Ku, C. C., Hsieh, M. S., and Lim, S. H.: The topographic effect in electromagnetic fields. Can. J.
Earth Sci. **10** 645-656, 1973.
16. Kumar, D., Singh, A., and Israil, M.: Necessity of Terrain Correction in Magnetotelluric Data
Recorded from Garhwal Himalayan Region, India. Geosciences.**11** 482, 2021.



17. Kumar, G. P., Manglik, A., and Thiagarajan, S.: Crustal Geoelectric Structure of the Sikkim
Himalaya and Adjoining Gangetic Foreland Basin. Tech. physics. **637** 238-250, 2014.
18. Kumar, S., Patro, P. K., and Chaudhary, B. S.: Three dimensional topography correction applied
to magnetotelluric data from Sikkim Himalayas. Physics Earth Planet. Int. **279** 33-46, 2018.
19. Kumar, S., Patro, K. P., and Chaudhary, B. S.: Subsurface Resistivity Image of Sikkim Himalaya
as Derived from Topography Corrected Magnetotelluric Data. Journal of the Geological Society
of India.  DOI: 10.1007/s12594-022-1985-2, 2022.
20. Larsen, J. C.: Removal of local surface conductivity effects from low frequency mantle response
curves, ActaGeodaet.,Geophys. et Montanist. Acad. Sci. Hung. Tomus. **12** (1-3) 183-186, 1977.
21. Mohan, K., Kumar, G. P., Chaudhary, P., Choudhary, V. K., Nagar, M., Khuswaha, D., Patel, P.,
Gandhi, D., and Rastogi, B. K.: Magnetotelluric Investigations to Identify Geothermal Source
Zone near ChabsarHotwater Spring Site, Ahmedabad, Gujarat, Northwest India. Geothermics.
**65** 198-209, 2017.
22. Nam, M. J., Kim, H. J., Song, Y., Lee, T. J., Son, J. S., and Suh, J. H.: Three-dimensional
topography corrections of magnetotelluric data. Geophysics J. Int. **174** 464-474, 2008.
23. Ngoc, P. V.: Magnetotelluric survey of the Mount Meager region of the Squamish Valley (British
Colombia). Geomagnetic Service of Canada, Earth Physics Branch of the Dept. of Energy,
Mines and Resources of Canada. Rep. 80-8-E, 1980.
24. Patro, P. K., and Harinarayana, T.: Deep Geoelectric Structure of the Sikkim Himalayas (NE
India) Using Magnetotelluric Studies. Phys. Earth Planet. Inter. **173** 171-176, 2009.
25. Patro, P. K.: Magnetotelluric Studies for Hydrocarbon and Geothermal Resources: Examples
from the Asian Region. Surveys Geophysics.**38** 1005-1041, 2017.
26. Rastogi, A.: A finite difference algorithm for two-dimensional inversion of geo-
electromagneticdata. Ph. D. Thesis, University of Roorkee (India), 1997.
27.  Rijo, L.: Modelling of electric and electromagnetic data: Ph.D. thesis, Univ. of Utah. **19**, 1977.
28. Suman, Tyagi, D. K., and Sherawat, R.: Topography distortion effect on Magnetotelluric (MT)
profiling of Sub-Himalayan region using two-dimensional modelling. J. Integr. Sci. Technol. **11**
462, 2023.

29. Thayer, R.E.: Topographic distortion of telluric currents: a simple calculation. Geophysics. **40** 91-
95, 1975.



30.  Tikhonov, A. N.: Determination of the Electrical Characteristics of the Deep Strata of the Earth's
Crust. DoklAkadamiaNauk.**73** 295-297, 1950.

31.  Tyagi, D. K.: 2D modeling and inversion of magnetotelluric data acquired in Garhwal Himalaya,
Ph. D. Thesis, 2007.

32.  Vozoff, K.: The magnetotelluric method, in Electromagnetic Methods in Applied Geophysics.ed.
Nabighian, M. N., Society of Exploration Geophysicists. **2** 641-711, 1991.

33.  Wannamaker, P. E., Stodt, J. A., Rijo, L.: Two-dimensional topographic responses in
magnetotellurics modeled using finite elements. Geophysics. **51** 2131-2144, 1986.

34.  Ward, S. H., Peeples, W. J., and Ryu, J.: Analysis of geo-electromagnetic data: Meth. Compo
Phys. **13** 163-238, 1973.

35.  Wescott, E. M., and Hessler, V. P.: The effect of topography and geology on telluric currents.
Jour. Geophysics. Res. **67** 4813-4823, 1962.

36.  Xiong, B., Luo, T. Y., Chen, L. W., Dai, S. K., Xu, Z. F., Li, C. W., Ding, Y. L., Wang, H. H.,
and Li, J. H.: Influence of Complex Topography on Magnetotelluric Observed Data Using
Three-Dimensional Numerical Simulation: A Case from Guangxi Area, China. Appl.
Geophysics. **17** 601-615, 2020.

37.  Zhang, K., Wei, W., Lu, Q., Dong, H., and Li, Y.: Theoretical Assessment of 3-D Magnetotelluric
Method for Oil and Gas Exploration: Synthetic Examples. J. Appl. Geophysics. **106** 23-36,
2014.