# Peer review of "Modeling of terrain effect in magnetotelluric data from Garhwal Himalaya Region"

_EGUsphere, 2023_

## Referee Comment (RC1)

**1 Modeling of terrain effect in magnetotelluric data from Garhwal Himalaya Region**

Suman Saini[1*], Deepak Kumar Tyagi[2], Sushil kumar[3], Rajeev Sehrawat[1**]

[1]Department of Physics, M. M. Engineering College, Maharishi Markandeshwar (Deemed to be)
University, Mullana-Ambala, Haryana, India 133207

Corresponding Author- Rajeev Sehrawat[1**]

**Email: rajeev.sehrawat@mmumullana.org

*Email: sumanabcd12@gmail.com

[2]Department of Physics, Krishnan College of Science and IT, M.J.P. Rohilkhand University, Bijnor,
Uttar Pradesh, India-246701

[3]Department of Geophysics, Kurukshetra University, Kurukshetra, India-136119

**ABSTRACT**

> I would start my abstract with an introductive sentence to explain what magnetotellurics is. For instance: "Magnetotelluric methods (MT) are passive geophysical techniques based on time variations of the geoelectric and geomagnetic fields in order to measure the electrical resistivity of subsurface layers." It will likely improve the readability of authors' manuscript and will make it more appealing.

[revised manuscript text omitted]

---

## Referee Comment (RC2)

Review of the manuscript "Modeling of terrain effect in magnetotelluric data from Garhwal Himalaya Region" by Suman Saini et. al., submitted to EGUsphere

**General Comments:**

The manuscript presents the modeling of the terrain effect in the MT data and compares the results from two different methods. Generally, the manuscript is organized well, and the content is suitable. However, several sections (e.g., section 4) are bad organized and need to be rewritten. The introduction was written in a very confusing style, especially in the second paragraph. The language is not fluent and precise, and there are many grammar errors and typos.

**Specific Comments:**

1. Line 16: Do not use the phrases such as "in this paper", "in this research paper" in the abstract.
2. Lines 20-21: This sentence is too long and a bit ambiguous, please reform it.
3. Lines 23-25: This sentence is a bit ambiguous and needs to be modified.

Section Introduction
4. Line 29: "time-varying measured components of earth's" may be deleted.
5. Line 30: Modify "interior" to "interior structures"
6. Line 43-44: What do the "different types of terrain geometrics" mean? Please clarify them.
7. Line 46: Please give at least three references to support the statement of "many researchers".
8. Lines 51-54: Finite difference method has also been used to treat the topographic effects, please search some papers. The Finite difference method has been mentioned as the main technique in the following paragraph, however, it is never been mentioned.
9. Line 59: What are "Two different terrain correction procedures"? One may have no idea about them until reading the corresponding contents in the references.

Section Methodology
10. This work compares two different techniques applied to remove the topography effect. Are there only two different techniques to treat the topography correction?
11. Lines 69-70: This sentence is confusing, please correct it.
12. Line 78: Were should be where.
13. Line 80: H- polarization may be replace with TM mode, which was used in previous section.
14. Line 85 and Line 101: "Where" should be use the lower case format.
15. Lines 101-102: This sentence is a bit confusing and needs to be modified.
16. Line 108: equation (7) needs to be corrected.
17. Lines 108-119: Equations should be described in detail.

Section Testing

18. Line 135: Figure 2, the symbol is too large to be seen clear. And a plot of relative error is better to be provide to show the similarity or the deviation.
19. Line 138:Fig.3 should be replaced with Figure 3.
20. Line 146: The symbol is too large to be seen clear in Fig. 3.
21. Lines 141-145: The results of the relative errors should be stated in detail.

Section Modeling

22. Lines 154-174: This section is only the construction of the model rather than the modeling. The content of the modeling should contain the model, gridding, method and results.

Section Result and Discussion

23. This section states the modeling results of three different models, and there seems no discussion in this section.

Section Conclusions

Line 244: what does the "afficacy" mean?

Lines 248-250: The last sentence is not the conclusion from this study. It is well known before this study.

---

## Author Comment (AC1)

**Response to the Reviewers comments**

**Comment** **1:** Iwouldstartmyabstractwithanintroductivesentencetoexplainwhatmagnetotelluric sis.Forinstance: "Magnetotelluric methods (MT) are passive geophysical techniques based on time variations of thegeoelectric and geomagnetic fields in order tomeasure the electrical resistivity of subsurface layers."It will likely improve thereadability ofauthors' manuscript and will make it more appealing.

**Ans 1:** The suggestion has been incorporated in the abstract of revised manuscript on page no. 1 and line 12-15. The abstract has been started with'Magnetotelluric methods (MT) are passive geophysical techniques based on time variations of the geoelectric and geomagnetic field in order to measure the electrical resistivity of surface layer. It is most effective geophysical techniques to study the deep structure of the Earth's crust, particularly in steep terrain like the Garhwal Himalaya region'.

**Comment 2:** In the paper I cannot find a section devoted to data and a list of publications or repository where I can find data is lacking, even though previous publications are cited throughout the manuscript.. Data sources MUST beadded explicitly at least at the end of the paper with a section called "DATAAVAILABILITY"

**Ans:** The data source has been added to the manuscript in model figure 1on page no. 7 and line 34.

**Comment 3:**I suggest: the shallow layers of the Earth

Ans:  The suggestion has been incorporated 'in the interior of earth' has been replaced by 'the shallow layers of the Earth 'on page no 2and line 32 in the revised manuscript.

**Comment 4:** To add the word 'Geometries'

Ans 4: Geometries has been added on page no 2 and line 46

**Comment 5:** Are distorsion coefficients complex numbe? If yes, authors can use this adjective,otherwise,please,removeit,otherwiseit can produce misunderstandings

Ans 5: The complex coefficients $D(f,x)$ are distortion coefficients has been added on page no. 4 and line 86 in the revised manuscript

**Comment 6:** Write the tensorial product in anappropriateway,please.

Ans 6: Also write equation no. 7 and equation 8 on page no. 6 and line 110 and 112

**Comment 7:** 10 kOhm . m

Ans 7: 10 kOm. M has been added on page no. 6 and line 124

**Comment 8:** The picture is not to scaleboth in the vertical andhorizontal axis. I ask theauthorstoimproveit.

Ans 8: Fig. 1: Topographic model of 500 Ω-m half-space with a resistive body of 10 kΩ.m embedded from the surface relief (Chouteau and Bouchard, 1988) has been added the scale on page no 7 and line 134

**Comment 9:** I suggest to set more appropriate vertical scales in order to allow betterreadabilityofthefigures.E.g.,fortherelativeerror(Res.)plot0-1.0instead of [-0.5, 2]

Ans 9: The vertical scale of fig 4 for the relative error (res) plot 0-1.0 has been fixed to have better readability in the revised manuscript.

**Comment 10:** Add a white space on line 163 page no 10.

Ans 10:The space has been added on page no. 10 and line 163 of the revised manuscript.

**Comment 11:**Please add: Figure not toscaleforbetterreadability.

Ans 11: Scale has been added to have better readabilityin figure 5 in line no 173 on page no. 11 in the revised manuscript.

**Comment 12:** Arealthesedigitssignificant?Please, putthe significantones. Thanks

Ans 12: All digits has been corrected up to significant figures on line numbers 82,87,90, 210,228,229, 230 in the revised manuscript.

**Comment 13:** Isuggesttochangethesettings fortheverticalaxistoimprove thequalityof thisfigure:there are several subplots, so,please, save space plotting data well distributed in each plot.

Ans 13: In order to save space vertical axis scale has been improved in fig 6 and fig 7 on page 12 ,13 and line 193 ,197 in the revised manuscript.

**Comment 14:** S. Saini I suppose.

Ans 14: Thankyou sir, yes, you are correct, S.Saini has been added on page no 18 and line 253 of the revised manuscript.

---

## Author Comment (AC2)

**Response to the Reviewers comments**

**Comment 1**: Do not use the phrases such as"in this paper","in this research paper"in the abstract.

**Ans 1:** has been added on page no 2 and line 46 in revised manuscript.

**Comment 2**: This sentence is tool on gand a bit ambiguous, please reform it.

**Ans 2:** The relative errors between two terrain correction procedures were calculated with respect to flat earth and were very less or almost zero for most of the sites along the Roorkee to Gangotri profile except at the foothill where the error was high at lower periods has been added on page no 1 and line 21-24 in revised manuscript.

**Comment 3**: Modify"interior"to"interior structures"

**Ans 3:** interior structures" has been added on page no 2 and line 31 in revised manuscript.

**Comment 4**: What do the"different types of terrain geometrics" mean? Please clarify them.

**Ans 4:** The 2D numerical techniques have been used for different type terrain geometries to remove topography effects from the data. The analogue, analytic, and numerical solution methods were used to study the analogue model (Wescott and Hessler, 1962; Faradzhev et al., 1972).Various two-dimensional (2D) numerical techniques have been used for the numerical treatment of the topographic effects like networking analogy (Ku et al., 1973; NgCo, 1980) and Rayleigh scattering numerical modeling techniques (Reddig and Jiracek, 1984; Jiracek et al., 1989), finite element method (Wannamaker Stodt and Rijo, 1986; Frankle et al., 2007) and finite difference method (Josef Pek and Tomas Verner, 1996; Yutaka Sasaki, 2003 and Tyagi et al., 2007) has been added on page no. 3 and lines 44-52 in revised manuscript.

**Comment 5**: Please give at least three references to support the statement of "many researchers".

**Ans 5:** references has been added on page no.3 and line 53-54 in revised manuscript.

**Comment 6**: Finite difference method has also been used to treat the topographic effects, please search some papers. The Finite difference method has been mentioned as the main technique in the following paragraph, however, it is never been mentioned.

**Ans 6:** finite difference method (Josef Pek and Tomas Verner, 1996; Yutaka Sasaki, 2003 and Tyagi et al., 2007; 2019) has been added on page no. 3 and lines 51-52 in revised manuscript.

**Comment 7**: This sentence is confusing, please correct it.

**Ans 7**: Two correction procedures, first adopted by Chouteau and Bouchard (1988) and second by Nam et al. (2008), were used to correct the MT data has been added on

page no.4 and line 71 and 72 in revised manuscript.

**Comment 8**: Were should be where.

**Ans 8:** where has been added on page no. 4 and line 80 in revised manuscript.

**Comment 9**: H-polarization may be replace with TM mode, which was used in previous section.

**Ans 9:** TM –mode has been added on page no 4 line 82 in revised manuscript.

**Comment 10**: "Where" should be use the lower case format.

**Ans 10:** where has been added on page no.4,5 and line 87, 94 and 102 in revised manuscript.

**Comment 11**: equation(7) needs to be corrected.

**Ans 11:** equatio (7) has been added on page no. 6 and line 110 in revised manuscript.

**Comment 12**: Figure2, the symbol is too large to be seen clear. An daplot of relative error is better to be provide to show the similarity or the deviation.

**Ans 12:** Figure 2, has been added on page no. 8 and line 137 in revised manuscript.

**Comment 13**:Fig.3 should be replaced with Figure3.

**Ans 13**: Figure 3 has been added on page no. 9 and line 149 in revised manuscript.

**Comment 14**: The symbol is too large to be seen clear inFig.3.

**Ans 14:** figure 3 has been added on page 9 and line 148 in revised manuscript.

**Comment 15**: This section is only the construction of the model rather an the modeling. The content of the modeling should contain the model, gridding, method and results.

**Ans 15:** The content of the modelling should contain the model, gridding, method, revised manuscript page no. 11 and lines 156–176, and result detail in the result and discussion.

**Comment 16** : what does the "afficacy" mean?

**Ans 16**: accuracy has been added on page no. 17 and line 246 in revised manuscript.

---

## Referee Report (RR1)

Review of the revised manuscript "Modeling of terrain effect in magnetotelluric data from Garhwal Himalaya Region" by Suman Saini et. al., submitted to EGUsphere

**General Comments:**

The revised manuscript has been improved compared to the initial submission. Most of the issues I concerned in the first review have been corrected or addressed. However, the authors didn't reply the comments point by point, so I have to examine the responses of comments. The lines addressed in the responses letter were not consistent with the actual positon in the manuscript.

In addition, the authors didn't reply several comments I gave in my first review, perhaps the authors believed the reviews were not suitable, even so they should classified or addressed the reasons in the responses letter. In my first review, 25 pieces of comments were given, however, the authors answered 16 points.

There are still some points needed to be corrected or modified before this manuscript can be accepted besides those comments not addressed in the first review

**Specific Comments:**

Line 12-14:"Magnetotelluric methods" should be written in singular format, and it is used to measure the electrical structure of the subsurface rather than surface.

Line 14-15: The sentence is written in a wrong style. It is a most effective geophysical technique.

Line 17: Do not use "In this paper" in the abstract, the abstract should be an entire presentation.

Line 22-24: The word "less" is misused. And this sentence is too long to understand, I recommend to use short sentence to represent the meanings.

Line 25:The abbreviation "TCR1" and "TCR2" should be used together with the full name when they are first mentioned, unless they are phrases known well.

Line 108: Where should be replaced with where.

Lines 107-116:The symbols used in the equations were not explained in detail.

Line 129: 10 k$\Omega$ m should be replaced with $10000$ $\Omega$ m.

Line 142:It will be better to add the results of Chouteau and Bouchard (1988) in Figure 2, so that the readers may understand the comparison in a very easy mean. Or the readers need to find the paper by Chouteau and Bouchard (1988) to compare the results. Additionally, the symbols in Figure 2 are too large to be seen clear.

Line 161:I still insist that the content of the section 4 is not compatible with the title. The content is the construction of the model, however, the modelling does not only consists of the model, the results should also be included in this section. I have addressed this point in my first review.

---

## Author Response (AR2)

**Response to the Reviewers comments-1**

**Comment1:** I would start my abstract with an introductive sentence to explain what magnetotellurics is. For instance: "Magnetotelluric methods (MT) are passive geophysical techniques based on time variations of the geoelectric and geomagnetic fields in order to measure the electrical resistivity of subsurface layers."It will likely improve the readability of authors' manuscript and will make it more appealing.

**Ans 1:** The suggestion has been incorporated in the abstract of revised manuscript on page no. 1 and line 13-16. The abstract has been started with 'Magnetotelluric methods (MT) are passive geophysical techniques based on time variations of the geoelectric and geomagnetic field in order to measure the electrical resistivity of surface layer. It is most effective geophysical techniques to study the deep structure of the Earth's crust, particularly in steep terrain like the Garhwal Himalaya region'.

**Comment 2:** In the paper I cannot find a section devoted to data and a list of publications or repository where I can find data is lacking, even though previous publications are cited throughout the manuscript. Data sources MUST be added explicitly at least at the end of the paper with a section called "DATAAVAILABILITY"

**Ans 2 :** The data source has been added to the manuscript in model figure 1on page no. 7 and line 140.

**Comment 3:** I suggest: the shallow layers of the Earth

**Ans 3:** The suggestion has been incorporated 'in the interior of earth' has been replaced by 'the shallow layers of the Earth 'on page no 2 and line 32 in the revised manuscript.

**Comment 4:** To add the word 'Geometries'

Ans 4: Geometries has been added on page no 2 and line 46

**Comment 5:** To add the symbol :

**Ans 5:** : The symbol ':' has been added on page no. 3 and line 67

**Comment 6:** with

Ans 6: 'to' has been replaced by 'with' on page 4 and line 71

**Comment 7:** by the

Ans: 'with has been replaced by 'by the' on page 5 and line 91

**Comment 8:** Are distortion coefficients complex number? If yes authors can use this adjective, otherwise, please, remove it, otherwise it can produce misunderstandings

**Ans 8:** The complex coefficients $D(f,x)$ are distortion coefficients has been added on page no. 5 and line 94 in the revised manuscript

**Comment 9:** Write the tensorial product in an appropriate and please.

**Ans 9:** Also write equation no. 7 and equation 8 on page no. 6 and line 116 and 118

**Comment 10:** 10000 $\Omega$.m change in 10 k$\Omega$.m and ' $\Omega$.m'

**Ans 10:** now 10000 $\Omega$.m has been replaced in 10 k$\Omega$.m on page no. 6 and line 130 and '$\Omega$-m' repaced by '$\Omega$.m' in all revised manuscript.

**Comment 11:** The picture is not to scale both in the vertical and horizontal axis. I ask the authors to improve it.

**Ans 11:** Fig. 1: Topographic model of 500 Ω-m half-space with a resistive body of 10 kΩ.m embedded from the surface relief (Chouteau and Bouchard, 1988) has been added the scale on page no 7 and line 140 in revised manuscript.

**Comment 12:** I suggest to set more appropriate vertical scales in order to allow better readability of the figures. E.g., for the relative error (Res.) plot0-1.0 instead of [-0.5, 2]

**Ans 12:** The vertical scale of fig 4 for the relative error (res) plot 0-1.0 has been fixed to have better readability on page no. 10 and line 158 in the revised manuscript.

**Comment 13:** Add a white space on line 163 page no 10.

**Ans 13:** The space has been added on page no. 10 and line 168 of the revised manuscript.

**Comment 14:** Please add: Figure not to scale for better readability.

**Ans 14:** Scale has been added to have better readability in figure 5 in line no 180 on page no. 11 in the revised manuscript.

**Comment 15:** Are all these digits significant? Please, report just the significant ones. Thanks

**Ans 15:** All digits has been corrected up to significant figures on line numbers 178, 179, 189, 190,194, 197, 216, 217, 235, 236, 237 on page no. 11, 12, 14, 17, in the revised manuscript.

**Comment 16:** I suggest to change the settings for the vertical axis to improve the quality of this figure: there are several subplots, so, please, save space plotting data well distributed in each plot.

**Ans 16:** In order to save space vertical axis scale has been improved in fig 6 and fig 7 on page 13, 14 and line 200, 205 in the revised manuscript.

**Comment 17:** S. Saini I suppose.

**Ans 17:** Thank you sir, yes, you are correct, S. Saini has been added on page no 19 and line 259 of the revised manuscript.

**Response to the Reviewers comments-2**

**Comment 1**: Do not use the phrases such as "in this paper", in this research paper" in the abstract.

**Ans 1:** 'In this paper' has been replaced by 'In this study' on page no 1 and line 18 in revised manuscript.

**Comment 2**: This sentence is too long and a bit ambiguous, please reform it.

**Ans 2:** The relative errors between two terrain correction procedures were calculated with respect to flat earth and were very less or almost zero for most of the sites

along the Roorkee to Gangotri profile except at the foothill where the error was high at lower periods has been added on page no 1 and line 22-25 in revised manuscript.

**Comment 3**: Modify "interior" to "interior structures"

**Ans 3:** 'interior' has been modified by 'the shallow layers of the Earth' added on page no 2 and line 32 in revised manuscript.

**Comment 4**: What do the "different types of terrain geometrics" mean? Please clarify them.

**Ans 4:** The 2D numerical techniques have been used for different type terrain geometries to remove topography effects from the data. The analogue, analytic, and numerical solution methods were used to study the analogue model (Wescott and Hessler, 1962; Faradzhev et al., 1972).Various two-dimensional (2D) numerical techniques have been used for the numerical treatment of the topographic effects like networking analogy (Ku et al., 1973; NgCo, 1980) and Rayleigh scattering numerical modeling techniques (Reddig and Jiracek, 1984; Jiracek et al., 1989), finite element method (Wannamaker Stodt and Rijo, 1986; Frankle et al., 2007) and finite difference method (Josef Pek and Tomas Verner, 1996; Yutaka Sasaki, 2003 and Tyagi et al., 2007) has been added on page no. 3 and lines 45-52 in revised manuscript.

**Comment 5**: Please give at least three references to support the statement of "many researchers".

**Ans 5:** References have been added on page no.3 and line 54 in revised manuscript.

**Comment 6**: Finite difference method has also been used to treat the topographic effects, please search some papers. The Finite difference method has been mentioned as the main technique in the following paragraph, however, it is never been mentioned.

**Ans 6:** finite difference method (Josef Pek and Tomas Verner, 1996; Yutaka Sasaki, 2003 and Tyagi et al., 2007) has been added on page no. 3 and lines 50-51 in revised manuscript.

**Comment 7**: This sentence is confusing, please correct it.

**Ans 7**: Two correction procedures, first adopted by Chouteau and Bouchard (1988) and second by Nam et al. (2008), were used to correct the MT data has been added on page no.4 and line 77 and 78 in revised manuscript.

**Comment 8**: Were should be where.

**Ans 8:** 'where' has been added on page no. 4 and line 86 in revised manuscript.

**Comment 9**: H-polarization may be replaced with TM mode, which was used in previous section.

**Ans 9:** 'TM mode' has been added on page no 4 line 88 in revised manuscript.

**Comment 10**: "Where" should be use the lower-case format.

**Ans 10:** 'where' has been added on page no.4, 5 and line 86, 93, 100 and 109 in revised manuscript.

**Comment 11**: equation (7) needs to be corrected.

**Ans 11:** 'equation (7) 'has been added on page no. 6 and line 116 in revised manuscript.

**Comment 12**: Figure2, the symbol is too large to be seen clear. And a plot of relative error is better to be provide to show the similarity or the deviation.

**Ans 12:** 'Figure 2' has been added on page no. 8 and line 143 in revised manuscript.

**Comment 13**: Fig.3 should be replaced with Figure 3.

**Ans 13**: 'Fig. 3' has been replaced by 'Figure 3.' on page no. 9 and line 146 and 'Fig. ' replaced by 'Figure' in all revised manuscript.

**Comment 14**: The symbol is too large to be seen clear in Fig.3.

**Ans 14:** figure 3 has been added on page 9 and line 154 in revised manuscript.

**Comment 15**: This section is only the construction of the model rather the modeling. The content of the modeling should contain the model, gridding, method and results.

**Ans 15:** The content of the modeling should contain the model, gridding, method, revised manuscript page no. 11 and lines 162–182, and result detail in the result and discussion.

**Comment 16** : what does the "afficacy" mean?

**Ans 16**: afficacy" has been replaced by accuracy on page no. 18 and line 251 in revised manuscript.

**Response to the Editor comments**

**Comment 1-**Equations (5) and (6): the dot product symbol (.) has to be shifted a bit up.

**Ans1-** The symbol'.' has been replaced with '·' in the equations (5) and (6) on page no. 5, 6 and line 108, 113

**Comment 2-** Fig. 1: a) Within the Fig. 1: change "$10^7$ Ω m" in "$10^7$ Ω ·m", change "10000 Ω-m" in "10kΩ ·m", change "500Ω.m" in "500Ω·m";
b) explain in the caption or in the text the meaning of letters A to I; c) in caption ofFig. 1: the dot product symbol (.) has to be shifted a bit up: change "500Ω.m" in "500Ω·m", change "10kΩ.m" in "10kΩ·m"

**Ans 2-** All changing has been in the revised manuscript page no. 7

**Comment 3-** Fig. 2, Fig. 3, Fig. 6, Fig. 8, Fig. 10: with reference to resistivity, in the y-axis label change "Ω-m" with "Ω·m"; with reference to phase, in the y-axis label change " ° " with "deg"
**Ans 3-** All these units have been uniformed in fig 2, fig 3, fig 6, fig 8 and fig 10 of revised manuscript.

**Comment4-** The "Ω.m" appear many (too many!) times in the text and within the figures: Change it in "Ω·m"
**Ans 3-**"Ω.m" has been substituted by "Ω·m" at all places in the revised manuscript.
**Comment 5-**Reference: "Chouteau, M., and Bouchard, K.: Two-dimensional terrain correction in magnetotelluric. Surveys Geophysics. 53 854-862, 1988"., has to be corrected in "Chouteau, M., and Bouchard, K.: Two-dimensional terrain correction in magnetotelluric surveys. Geophysics. 53 854-862, 1988."

**Ans 5-** The suggestion has been incorporated in reference no 3" Chouteau, M., and Bouchard, K.: Two-dimensional terrain correction in magnetotelluric Surveys. Geophysics. **53** 854-862, 1988." In revised manuscript